

# An insight into impact of nanomaterials toxicity on human health

Wajhul Qamar[1], Shweta Gulia[2], Mohammad Athar[3,4], Razi Ahmad[5], Mohammad Tarique Imam[6], Prakash Chandra[2], Bhupendra Pratap Singh[7], Rizwanul Haque[8], Md. Imtaiyaz Hassan[9] and Shakilur Rahman[10]

[1] Department of Pharmacology and Toxicology and Central Laboratory, College of Pharmacy, King Saud University, Riyadh, Saudi Arabia
[2] Department of Biotechnology, Delhi Technological University, New Delhi, India
[3] Department of Medical Genetics, Umm Al-Qura University, Makkah, Saudi Arabia
[4] Science and Technology Unit, Umm Al-Qura University, Makkah, Saudi Arabia
[5] Department of Chemistry, Indian Institute of Technology, Delhi, New Delhi, India
[6] Department of Clinical Pharmacy, College of Pharmacy, Prince Sattam Bin Abdulaziz University, Al Kharj, Saudi Arabia
[7] Department of Environmental Studies, Deshbandhu College, University of Delhi, New Delhi, India
[8] Department of Biotechnology, Central University of South Bihar, Gaya, Bihar, India
[9] Centre for Interdisciplinary Research in Basic Sciences, Jamia Millia Islamia University, New Delhi, India
[10] Department of Medical Elementology and Toxicology, Jamia Hamdard University, New Delhi, India

Corresponding author
Shakilur Rahman, rshakilur25@gmail.com

## ABSTRACT

In recent years, advances in nanotechnology have significantly influenced electronics manufacturing, industrial processes, and medical research. Various industries have seen a surge in the use of nanomaterials. However, several researchers have raised the alarm about the toxicological nature of nanomaterials, which appear to be quite different from their crude forms. This altered nature can be attributed to their unique physicochemical profile. They can adversely affect human health and the environment. Nanomaterials that have been released into the environment tend to accumulate over time and can cause a significant impact on the ecosystem and organisms with adverse health effects. Increased use of nanoparticles has led to increased human exposure in their daily lives, making them more vulnerable to nanoparticle toxicity. Because of their small size, nanomaterials can readily cross biological membranes and enter cells, tissues, and organs. Therefore, the effect of nanomaterials on the human environment is of particular concern. The toxicological effects of nanomaterials and their mechanisms of action are being researched worldwide. Technological advances also support monitoring new nanomaterials marketed for industrial and household purposes. It is a challenging area because of the exceptional physicochemical properties of nanomaterials. This updated review focuses on the diverse toxicological perspective of nanomaterials. We have discussed the use of different types of nanoparticles and their physiochemical properties responsible for toxicity, routes of exposure, bio-distribution, and mechanism of toxicity. The review also includes various *in vivo* and *in vitro* methods of assessing the toxicity of nanomaterials. Finally, this review will provide a detailed insight into nano

material-induced toxicological response, which can be beneficial in designing safe and effective nanoparticles.

# INTRODUCTION

Toxicology is a multidimensional science branch involving the interaction of biological organisms, chemicals, and other agents to detect possible hazards (*Krewski et al., 2020*). The toxicity or toxic insult can be defined as any adverse biological effects of a product, irrespective of its origin. Many chemicals have been studied, and their toxicity and impact mechanisms have been published over several decades (*Carlin et al., 2015*; *Genuis & Kyrillos, 2017*; *Jaishankar et al., 2014*; *Lanphear, 2017*). Several studies have documented toxic events caused by nanomaterials over the past decade (*Barabadi et al., 2019*; *Chen et al., 2020*; *Ferdous & Nemmar, 2020*). If we accept the declaration of the father of toxicology, Paracelsus (1493–1541), that "All substances are poisons; there is none that is not a poison. The right dose differentiates a poison from a remedy", the nano-sized materials (nanoparticles, nanomaterials) are not an exception to this statement. In the case of nanomaterials, the surface area of the nanoparticles, relative to their general counterparts, intensifies their effects even at lower doses (*Ahmad, Khatoon & Sardar, 2014a*; *Ahmad et al., 2015*; *Ahmad & Sardar, 2015b*; *Khatoon, Ahmad & Sardar, 2015*; *Sardar & Ahmad, 2016*; *Sardar, Mishra & Ahmad, 2014*). Under the aegis of 'nanotoxicology,' the possible toxicological effects of nanomaterials are being examined. Nanomaterials possess unique catalytic, mechanical, and optical properties and electrical conductivity mainly due to their size in nanometers (*Ahmad et al., 2023*; *Ahmad & Sardar, 2015a*; *Ghosh et al., 2021a*, *2021b*; *Goel et al., 2023*; *Khan, Saeed & Khan, 2019*).

The branch of nanotoxicology may seem new, but it contains the same basic toxicological concepts. The instruments and techniques for the analysis can vary from the conventional ones, depending on the nature of the substance (nanomaterials in this case). The associated health issues are also growing with the increasing demand for nanomaterials in different industries and household goods. Toxicological approaches play a crucial role in regulating numerous substances and products' impacts on human and environmental health (*Malakar et al., 2021*; *Singh, Kumar & Jain, 2021*; *Stark et al., 2015*).

Nano-sized materials have shown benefits over their crude forms, mainly in formulation development. There are now recorded formulations containing one or more nanomaterials in many cosmetic products, pharmaceuticals, paints, pigments, *etc*. For example, nano-sized titanium dioxide ($TiO_2$) is used in biocatalytic processes and cosmetics (*Ahmad et al., 2019a*; *Ahmad, Khatoon & Sardar, 2013*), silver nanoparticles are used for their antibacterial in making stain and odor-resistant clothing (*Montes-Hernandez et al., 2021*; *Rajeshkumar et al., 2021*).
Nanotoxicology can address all the health and environmental issues raised by nano-sized materials. The physicochemical causes, exposure paths, biodistribution, molecular determinants, genotoxicity, and regulatory aspects are incorporated into nanotoxicology (*Arora, Rajwade & Paknikar, 2012*; *Goel et al., 2021*). The toxicity of nanomaterials depends on the interaction of their physicochemical assets with an organism's cellular and molecular components. Nanomaterials are highly reactive and may be harmful when interacting with biological systems and the environment. This can be attributed to quantum size effects and enormous surface area to volume ratio (*Ganguly, Breen & Pillai, 2018*; *Sahu & Hayes, 2017*). Several studies have revealed that nanoparticles (NPs), after intake, spread to several organs, including the liver, heart, spleen, brain, lungs, and gastrointestinal system, leading to adverse health effects (*Almeida et al., 2011*; *Bahadar et al., 2016*; *Khlebtsov & Dykman, 2011*).

This review aimed to get deeper insights into the toxicity profiling of nanomaterials and their consequential impacts on human health. It outlines the different classes of nanomaterials and their physiochemical properties responsible for toxicity. It discusses some commonly used nanoparticles and their associated adverse effects. It also discusses the possible routes of exposure (for example-respiratory tract, skin, and gastrointestinal tract) and its biodistribution. Furthermore, it discusses the detailed toxicological perspective with genotoxicity, immunotoxicity, oxidative stress, and inflammation as a mechanism of cellular toxicity. The discussion desk also discusses an entire section on the methods for assessing the toxicity of nanomaterials. Importantly, the present review summarizes the concept behind the toxicology of nano-sized materials and includes the advancements in nanotoxicology research and methods during the past two decades.

## The rationale of the study

The growing usage of nano-sized materials in various sectors and applications drove this study on nanotoxicology and its effects on human health. Although nanoparticles have many advantages, there are concerns about their possible negative impacts on human health and the environment. To ensure the safe and responsible use of these materials, it is necessary to recognize the potential hazards involved. The present study seeks to offer more understanding of nanomaterials' toxicity profile. Researchers can comprehend the underlying processes of toxicity by researching the physicochemical features of nanomaterials and their effects on cellular pathways. Nanotoxicology investigates the toxicity of nanomaterials and their interactions with biological systems to address these issues.

As nanotechnology develops, regulatory considerations are also crucial. Establishing suitable regulatory standards for the responsible use and disposal of nanomaterials is essential. This review raises these issues along with the toxicity of nanomaterials and its evaluation. It is intended to fill the gaps in concepts and understanding of nanotoxicology for students and researchers.

Overall, this study advances the science of nanotoxicology by shedding important light on the possible health risks of nanomaterials. This study provides decision-making information and aids in risk mitigation by knowing their toxicity and taking regulatory

considerations, offering the safe application of nanotechnology to benefit human health and the environment.

## Search methodology

This study used a survey and search approach to examine nanotoxicology's effects on human health. The subsequent actions were taken:

Review of the literature: Using particular keywords relating to nanotoxicology, nanomaterials, toxicity profiling, human health, and environmental consequences, relevant scientific literature was carefully examined from databases including PubMed, Scopus, and Web of Science. Articles were chosen based on predefined criteria, which included an emphasis on molecular components, cellular routes, physicochemical reasons, exposure pathways, biodistribution, genotoxicity, and regulatory issues.

Extraction of key data from chosen article on the physicochemical characteristics of nanomaterials, experimental procedures, toxicity assessment methods, molecular determinants, and regulatory issues.

Data were thematically analyzed to spot patterns, trends, and knowledge gaps in nanotoxicology. The synthesis of the findings provided a thorough understanding of the issue.

Interpretation and conclusion: The data were analyzed to reach relevant findings, respond to study-related questions, and better understand the toxicity of nanomaterials and their effects on human health. This study used a survey and search technique to acquire pertinent data, pinpoint knowledge gaps, and present a thorough analysis of nanotoxicology and its consequences for human health.

# CLASSES OF NANOMATERIALS

Depending on their practical application, nanomaterials can be classified based on size, morphology, state, and chemical composition. Generally, nanomaterials (NMs) are ranked based on their dimensionality, morphology, state, and chemical content (*Joudeh & Linke, 2022*; *Saleh, 2020*). This classification is also based on their size, which varies between 1–100 nm in at least one dimension.

## Classification of nanomaterials based on dimensions, morphology and state

There are numerous structural, dimensional, morphological, and compositional measures based on which nanomaterials are classified. They provide unique properties to nanomaterials and contribute to their fate and toxicity in the environment and human health (*Pokropivny & Skorokhod, 2007*; *Saleh, 2020*).

Nanomaterials are further distributed into four classes based on shape and dimensions. Zero-dimensional (0D) nanomaterials have all their dimensions <100 nm. These include nanomaterials shaped as cubes, spheres, polygons, nanorods, and hollow and nanomaterials existing as quantum dots (QDs). One-dimensional nanomaterials have only one dimension on the nanoscale. Examples of 1D nanomaterials are nanorods, nanotubes, nanofibers, nanowires, and metallics. Two-dimensional nanomaterials have two

dimensions in the nanoscale, including thin films, nanoplates, and so on, which may be in single or double layers. 2D nanomaterials can exist in crystalline or amorphous forms. 3D nanomaterials have more than two dimensions. Nanotubes, fullerenes, honeycombs, and fibers are a few examples of 3D nanomaterials (*Aversa et al., 2018*; *Sefadi & Mochane, 2020*; *Shiau et al., 2018*). Morphologically, nanoparticles can be distributed based on their formation as flat spheres and the aspect ratio (High and low), including nano zigzags, nanopillars, nanospheres, nanopyramids, *etc*. Another class of nanomaterials is centered on the states in which they can exist, such as suspension, colloid, or dispersed, *e.g.*, magnetic nanoparticles. Elements of nanomaterials are an essential factor in categorizing them (*Saleh, 2020*).

## Classification of nanomaterials-based on the chemical composition

The largest category of nanomaterials is based on chemical composition and constitutes single constituents, composite, and inorganic and organic nanomaterials (*Saleh, 2020*). This class comprises nanomaterials like composites, carbonaceous, metallic, metallic oxide, polymeric, *etc*., (*Mekuye & Abera, 2023*; *Saleh, 2020*). Four types of nanomaterials categories are based on chemical composition: carbon-based nanomaterials, inorganic-based nanomaterials, organic-based nanomaterials, and composite-based nanomaterials (*Majhi & Yadav, 2021*). Carbon is the core constituent of carbonaceous nanomaterials, while metallic nanomaterials are differentiated based on the metals they are made from. Carbon-based nanomaterials include graphene, fullerene, single-walled carbon nanotube, multi-walled carbon nanotube, carbon fiber, activated carbon, and carbon black. The metals forming metallic nanomaterials are usually Cu, Ag, Al, Zn, Fe, *etc*., thus having catalytic and adsorptive properties (*Kim & Lee, 2018*; *Li et al., 2019*; *Vijayakumar et al., 2022*). Through certain processes like hydrothermal, doping, or sol-gel reactions, metallic oxide nanoparticles (*e.g.*, $TiO_2$, $Fe_2O_3$, and $SiO_2$) can be produced. This class of nanomaterials comes with additional applications, like sensors, semiconductors, *etc*., (*Saleh & Fadillah, 2019*). The organic-based nanomaterials are formed from organic materials, excluding carbon materials, for instance, dendrimers, cyclodextrin, liposomes, and micelle. Next, bimetallic nanomaterials are a combination of metals with different properties, such as Ag-Cu and Fe-Cu, which are further classified based on their structure (*Hao et al., 2020*; *Lozhkomoev et al., 2019*; *Saleh & Fadillah, 2019*). Other nanomaterials based on composition are branched dendrimers, ceramic nanomaterials, nanogels, core-shell nanomaterials, and polymeric nanomaterials (*Saleh, 2020*). A class of nanomaterials is designed for drug delivery known as lipid-based nanomaterials (*García-Pinel et al., 2019*). These nanomaterials could target certain hydrophilic and hydrophobic molecules in the human body and are relatively stable, less toxic, and specific, *e.g.*, nanostructure lipid carriers, solid lipid nanoparticles, and liposomes. Liposomes are mainly formed of cholesterol and phospholipid compounds (*García-Pinel et al., 2019*; *Zhong & Zhang, 2019*). Nanomaterials also occur as quantum dots and can absorb UV light and white light and reemit them at certain wavelengths. Hence, quantum dots come with exclusive optical properties and electronic ones (*Alavi, Jabari & Jabbari, 2021*).

# PHYSICOCHEMICAL PROPERTIES OF NANOMATERIALS RESPONSIBLE FOR TOXICITY

Studying nanomaterials from the physicochemical angle is essential for understanding the toxic effect. The physicochemical properties are responsible for the interaction between the nanoparticle and the target molecule or cell (Chandra, Kim & Rhee, 2013). These properties include particle shape, size, composition, stability, structural dimensions, concentration, nanoparticle morphology, and surface properties (area, roughness, energy, charge), including functional groups. These interactions also determine the entry of nanoparticles inside the cellular pathways, their translocation, exposure, and further interactions with the molecules and entities inside the cell. Such interactions can be exemplified as hydrophobic, electrostatic, steric, solvent, and biological interactions. There are various mechanisms of nanoparticle toxicity. These include the release of more reactive ionic form from nanoparticle surface, ROS generation, lipid peroxidation, protein denaturation, inflammation, endothelial dysfunction, mitochondrial perturbation, phagocytic function impairment, and altered cell cycle regulation (Gupta & Xie, 2018). An *in vivo* study conducted by Zhang et al. (2011) demonstrates size-dependent nanoparticle toxicity, indicating that 10 and 60 nm PEG-coated gold nanoparticles increased alanine transaminase and aspartate transaminase levels resulting in liver injury. In an *in vitro* experiment conducted by Ng et al. (2017) MRC5 lung cells treated with ZnO NP-treated released a substantial amount of extracellular lactate dehydrogenase and had lower cell viability, indicating cellular damage and cytotoxicity.

## Effects of physicochemical properties of nanomaterials on a cellular level

All the properties mentioned above are responsible for the toxicity displayed by nanomaterials. Several research studies using different cell types have examined toxicity effects due to the physicochemical properties of nanoparticles and engineered nanomaterials. One such study reported the toxic effect of graphene family nanomaterials on ocular cells and the possible risk of applying graphene family nanomaterials in biomedical (Borandeh et al., 2021). Graphene family nanomaterials' use has been proposed in ocular drugs, contact lenses, and ocular drug delivery due to their large π-conjugated aromatic structure and specific surface area for example, graphene oxide and reduced graphene oxide are analogs of graphene family nanomaterial and effect cell viability (Borandeh et al., 2021; Ge et al., 2018; Oliveira et al., 2022). Graphene oxide nanoplates with 11 nm dimensions can enter the human mesenchymal stem cell nucleus more quickly and cause genotoxicity than a graphene oxide nanoplate with a 3 μ dimension. Also, nanoplates higher in concentration can cause more toxicity than those at low concentrations, as stated in studies (Borandeh et al., 2021; Ge et al., 2018; Oliveira et al., 2022). Reduced graphene oxide nanoplates of micron size were highly toxic at a 100 μg/ml concentration.

In contrast, the smaller-sized reactive graphene oxide nanoplate (11 nm), at a concentration as low as 0.1–1 μg/ml, caused genotoxicity on translocation to the nucleus
(*Akhavan, Ghaderi & Akhavan, 2012*). The shape of the nanomaterial is another nanotoxicity-determining factor in living cells. In another study about the effect and mechanism of nanotoxicity, it was concluded that there was growth inhibition and apoptosis in primary rat osteoblasts by hydroxyapatite nanoparticles which are needle-shaped and short. At the same time, this was less likely in the case of nanoparticles, having spherical, long rod shapes (*Jain & Patel, 2021*; *Xue, Wu & Sun, 2012*). The chemical composition of nanomaterials is also a nanotoxicity causal factor. Particle dissolution, for example, can generate toxicity imparting ionic species. Nanotoxicity of nanoparticles like $CuO$, $CdO$, and $TiO_2$ was found to cause DNA damage, which was high in the case of $CuO$ but comparatively lower in $TiO$ and $CdO$ (*Franklin et al., 2007*; *Zhu et al., 2013*). Table 1 depicts a few examples of nanomaterials' size, shape, and concentration-dependent toxicity in mammalian cells.

## Effects of physicochemical properties of nanomaterials on the environment

The properties of nanomaterials that are accountable for nanotoxicity among living cells are the source of nanotoxicity in the environment. These factors can affect microorganisms, plants, soil quality, water quality, quality of air, crop production, pollution levels, and aquatic life. To simplify this physical and chemical effect, it should be first understood that when nanomaterials encounter water, atmosphere, or soil, a corona forms on their outer surface. Thus, the nanomaterials interact with their corona and molecules present on the cell wall and membrane (*Docter et al., 2015*; *Foroozandeh & Aziz, 2015*; *Juárez-Maldonado et al., 2021*; *Nel et al., 2009*). Consequently, conditional to the corona composition, nanomaterials with the same physical and chemical properties and the same corona composition but variable properties will unlikely affect a cell or microorganism. Table 2 presents a few examples of nanomaterials and their physicochemical properties affecting mammalian cells and their associated mechanism. Although it hasn't been understood how nanomaterials produce toxicity in flora and microorganisms, reactive oxygen species (ROS) generation can stimulate specific defense mechanisms in cells or microorganisms that can lead to cell death (*Smerkova et al., 2020*; *Zhao et al., 2020*). However, ROS generation is not the only mechanism that can cause cytotoxicity, and antimicrobial molecule Reactive nitrogen species (RNS) generation enabled by nanomaterials can also cause extreme levels of cellular stress that can lead to toxicity (*Balážová, Baláž & Babula, 2020*; *Juárez-Maldonado et al., 2021*; *Wang et al., 2020b*; *Zhao et al., 2020*). Several nanoparticles are known for inducing cytotoxicity and inhibition in soil microorganisms by acting as potent antimicrobial agents, and cytotoxicity depends on the concentration of nanomaterials to which the microbe is exposed (*Abdulla et al., 2021*; *Kumari et al., 2014*). A higher concentration of nanomaterials has been reported to cause DNA damage, lipid peroxidation, ROS generation, ion release, ATP depletion, and cell damage in soil microflora. The toxicity of nanomaterials towards microorganisms also depends upon the type and nature of the nanomaterial. Nanoparticles metallic in nature such as $ZnO$, $CuO$, $Ag$, $CeO_2$, or $Fe_3O_4$ nanoparticles could modify the composition of soil microflora (*McKee & Filser, 2016*).

**Table 1 Nanomaterials and their toxicity on mammalian cells.**

| S. no. | Nanoparticle type/ nanomaterial | Size/shape/concentration of nanomaterial | Effect on mammalian cells | References |
|---|---|---|---|---|
| 1. | Silver | 20 µm | Cytotoxic to murine macrophage cell line has been found in the colon and blood of patients with colon cancer and blood cancer, respectively | Gatti (2004), Jain et al. (2011), Soto et al. (2005), Takenaka et al. (2001) |
| 2. | Superparamagnetic Iron oxide | Concentration >100 µg/ml | Reportedly, impairs the functions of DNA, nucleus and mitochondria and causes inflammation | Brunner et al. (2006), Nel et al. (2006), Oberdörster, Stone & Donaldson (2007), Singh et al. (2010) Sudhakar et al. (2021) |
| 3. | Zirconium dioxide | 11 nm particle size | Increases viral receptor expressions | Lucarelli et al. (2004) |
| 4. | Cadmium nanorod | 30–50 nm diameter & 50–1,100 nm length, concentration 1,000–10,000 mg/kg. | Genotoxicity, oxidative stress and DNA damage in Kunming mice | Demir (2021), Liu et al. (2014a) |
| 5. | Cerium oxide nanorods | Length >200 nm | Progressive pro-inflammatory effect and cytotoxicity in Human myeloid cell line | Ji et al. (2012) |
| 6. | Hydroxyapatite (nano-HAP) | Crystals, H-rod, H-needle, H-sphere, H-plate | Decreased cell viability and consequent necrosis in rat aortic smooth muscle cells | Huang, Sun & Ouyang (2019) |
| 7. | Citrate capped gold nanoparticle | 5 nm | Toxicity, increased Cytokine production in mouse fibroblast | Jia et al. (2017) |
| 8. | Nickel nanowire | 33 nm diameter, 5.4 µm length, 5 µg/ml | Cytotoxicity and decreased cell viability in human colorectal carcinoma HCT 116 cells | Perez et al. (2016) |
| 9. | Zinc oxide nanoparticle | 4–20 nm diameter | Low viability, ROS production, cytotoxicity in human immune cells (e.g., monocyte) | Hanley et al. (2009) |
| 10. | Titanium oxide | 70 nm diameter, 50 µg/ml | Inflammation, elevated IL-8 in human microvascular endothelial cells | Peters et al. (2004) |
| | | 3 and 600 µg/ml | Shrinking of cells, lower metabolic activity, releasing of LDH,ROS production in mouse fibroblastL929 | Jin et al. (2008) |
| 11. | Graphene oxide | Upton 25 µg/ml | Effected antigen inhibition linked to downregulated intracellular levels of immune proteasome | Seabra et al. (2014), Tkach et al. (2013) |
| 12. | Mesoporous silica nanoparticle | 100 nm | Membrane deformities and hemolysis in human red blood cells | Lin & Haynes (2010) |

Some nanoparticles like carbon nanoparticles are known to be more toxic to microbes in soil (Chen et al., 2019). Nanomaterials have a variable effect on distinct microorganisms depending on their concentration, size, or parent material. In a research study based on the effect of a few engineered nanomaterials on two different organisms, it was deduced that the growth inhibition of both microbes varied when subjected to unrelated nanomaterials with different concentrations. Nanomaterials such as ZnO (4 mg L$^{-1}$), In$_2$O$_3$, γ-Al$_2$O$_3$, and TiO$_2$ inhibited the growth in Skeletonoma Costatumin up to 100% (with ZnO).

In comparison, in the case of nanomaterials like α-Al$_2$O$_3$ and SnO$_2$ (1,280 mg L$^{-1}$), inhibition did not occur fully, being 35% in the case of α-Al$_2$O$_3$ and 24% with SnO$_2$ (Wong et al., 2020). As a matter of size, shape, material, and coating, it has been described that increased concentration of nanomaterials to a certain level at which it becomes toxic and

**Table 2 Physico-chemical properties of nanomaterial responsible for its cytotoxicity and associated mechanism.**

| S. no. | Type of nanoparticle | Source of nanomaterial | Physico-chemical property responsible for cytotoxicity | Mechanisms for cytotoxic effect | References |
|---|---|---|---|---|---|
| 1. | Mesoporous silica nanoparticles (MSNs) | Tetra ethyl orthosilicate | Size (~600 nm) dependent cytotoxicity | Severe local membrane distortion resulting in RBC spherocytosis, internalization of the particles, and ultimately hemolysis. | *Zhao et al. (2011)* |
| 2. | Graphene oxide (GO) nanosheets | Oxidation of graphite oxide | Concentration-dependent cytotoxicity | Direct interactions between the cell membrane and GO nanosheets that cause physical harm to the cell membrane cause cytotoxicity due to high absorption | *Hu et al. (2011)*, *Zhu et al. (2020)* |
| 3. | Titanium dioxide (TiO$_2$) nanoparticles | Titanium dioxide | Concentration (100 µg ml$^{-1}$) dependent neurotoxicity | Tau-TiO$_2$ NP interaction is observed when neuroblastoma cells are exposed to the nanoparticles. Microtubules may become unstable as a result of interactions between TiO$_2$ and tau proteins, tubule heterodimers, and microtubules, which increases the neurotoxicity of TiO$_2$ NPs | *Mao et al. (2015)*, *Sukhanova et al. (2018)* |
| 4. | Iron oxide nanoparticles (IONPs) | Iron oxide | Citrate and dextran coating on the surface of NP. | Highly reactive hydroxyl radicals are produced by the Haber-Weiss reaction in cells (*e.g.*, HUVECs). At larger intracellular quantities of iron salts, the Fenton reaction is detectable. The free radicals have the potential to harm DNA, cell membranes, the cytoskeleton, and ECM. They can also directly or indirectly mediate signal transduction pathways through bioactive mediators | *Puppi et al. (2011)*, *Sadaf et al. (2020)*, *Sukhanova et al. (2018)*, *Wu et al. (2010)* |
| 5. | Au55 clusters | Reduction of (triphenylphosphine)gold chloride with diborane followed by incorporation into SBA-15 silica mesopores | Cuboctahedral structure and 1.4 nm size | Cytotoxicity (necrosis) brought on by the remarkably strong interaction of the 1.4 nm particles with the main grooves of DNA | *Schmid (2008)*, *Schmid et al. (2008)*. |
| 6. | Cadmium telluride quantum dot NPs | 1-thioglycerol, L-cysteine, thioglycolic acid and Cd$^{2+}$ | Chemical compostion-presence of cadmium and tellurium | CdTe-QDs alter the potential of the mitochondrial membrane *via* increasing Ca$^{2+}$ levels. The cadmium component of the NPs causes CdTe-QDs' impact on Ca$^{2+}$. Increased intracellular Ca$^{2+}$ has been linked to cadmium-induced apoptosis | *Belyaeva et al. (2012)*, *Nguyen et al. (2015)* |
| 7. | Polystyrene NPs | n-pentanol, sodium dodecyl sulphate, ammonium persulfate | Surface charge | Positively charged nanoparticles (NPs) more efficiently cross the membrane, but also because they are more tightly linked to the negatively charged DNA, damaging it and lengthening the G0/G1 phase of the cell cycle as a result | *Liu et al. (2011)*, *Sukhanova et al. (2018)* |

| S. no. | Type of nanoparticle | Source of nanomaterial | Physico-chemical property responsible for cytotoxicity | Mechanisms for cytotoxic effect | References |
|---|---|---|---|---|---|
| | | Table 2 (continued) | | | |
| 8. | Casein-coated iron oxide nanoparticles | Casein, iron chloride tetrahydrate, and iron chloride hexahydrate | NP shell | Shell enhances durability and guards NP against desalination and oxidative or photolytic deterioration. As a result, the toxicity of NP is reduced. Due to the wide range of modifiers, altering the NP characteristics as well as their particular transport and accumulation is possible that can increase or decrease the toxicity | Huang et al. (2013) |
| 9. | Zinc oxide NP | Zinc salts | $Zn^{2+}$ ions | ZnO NPs generate an increase in cytoplasmic and mitochondrial $Zn^{2+}$ levels, which can ultimately result in mitochondrial damage and apoptosis in cells | Soenen et al. (2015), Wilhelmi et al. (2013) |
| 10. | Gold NP | $Au^{+1}$ or $Au^{3+}$ | Plasmonic properties | The cytotoxicity is caused by the ionic interactions of the Au-NPs with the plasma membrane. Genes including those connected to the cell cycle, particularly those engaged in the G1 phase and those involved in the nucleic metabolic process, are down-regulated | Lee et al. (2019) |
| 11. | Polystyrene latex NP | Styrene, sodium hydrogen carbonate and potassium persulfate | Surface modification and particle size (50 and 100 nm) | On cell membranes, there have been reports of significant damage and holes that have not been seen with other kinds of NPs. Along with enhanced CXCL8 production, this also induces apoptotic (caspase-3/7 and caspase-9) cell death (IL-8) and cell detachment | Ruenraroengsak et al. (2012) |

bio-stimulatory activity starts decreasing (Agathokleous et al., 2019). Similarly, nanomaterials bigger in size have less effect on cells (Ball, 2017; Klaine et al., 2008).

## COMMONLY USED NANOMATERIAL AND THEIR ADVERSE HEALTH EFFECTS

NPs (with a diameter of 100 nm or less) are found in the air and some workplaces. NPs can be either naturally occurring (like dust, protein molecules, viruses, aerosol, mineralized natural materials, volcanic ash, etc.) or artificially produced (like titanium dioxide, carbon black, carbon nanotubes, silver, zinc, and copper oxide) (Alshammari et al., 2023; Cho et al., 2019; Mekuye & Abera, 2023). Engineered nanomaterials have exceptionally large surface areas and high percentages of their component atoms on the surface because they have at least one dimension smaller than 100 nm. Many nanoparticles acquire exceptional

reactivity as a result, which opens up new avenues for their application in electronics, medicine, environmental cleanup, catalysis, and consumer goods. Concern regarding the possible hazardous effects of nanoparticles due to their usage or inadvertent release into the environment has grown in tandem with the growing interest in the advantages they may provide (*Kabir et al., 2018*; *Martínez et al., 2020*; *Ray, Yu & Fu, 2009*).

## Toxicity of metal-based nanoparticles

The elements gold, silver, copper, nickel, cobalt, zinc, and titanium dioxide have all been used to create nanoparticles. A significant portion of the expanding nanotechnology business is made up of metal and metal oxide nanoparticles. The use of metallic nanoparticles is rising, and with that comes the risk that these particles will be released into the environment.

As well as being utilized as a bactericide, silver nanoparticles have found use in the production of stain-and odor-resistant textiles, sensors, inks, and catalysts (*Baker et al., 2005*). Antimicrobial products using silver nanoparticles are becoming increasingly popular in the Western hemisphere. More and more research points to silver nanoparticles' extreme toxicity to mammalian cells (*Ferdous & Nemmar, 2020*; *Rohde et al., 2021*). Brain, liver, and stem cells are all vulnerable to the toxicity of silver nanoparticles (*Ferdous & Nemmar, 2020*). Bactericidal treatments made with nanocopper are increasingly popular (*Ermini & Voliani, 2021*). Coatings and sealants with these additives have better thermal and electrical conductivity and may be used to filter air and liquids. They can also be applied to integrated circuits and batteries. Several aquatic species, including vertebrates and invertebrates, have had it shown that copper and silver nanoparticles are extremely hazardous to them (*Das, Xenopoulos & Metcalfe, 2013*; *Dube & Okuthe, 2023*). TiO$_2$ nanoparticles are a manufactured nanoparticle with widespread application in food colorings, sunscreens, and cosmetics. Because of their great stability, anti-corrosion, and photocatalytic capabilities, titanium dioxide nanoparticles (TiO$_2$-NPs) were mass-produced and employed extensively. It is one of the most extensively utilized nanomaterials and has lately found use in several agricultural industries. Emerging data suggest that TiO$_2$ nanoparticles can cause inflammatory and genotoxic reactions in various animal and human cell lines. DNA damage is caused by the high levels of hydroxyl free radical produced by TiO$_2$ nanoparticles (*Chen, Yan & Li, 2014*; *Suzuki et al., 2020*). Various mechanisms and properties through which nanoparticles exert toxicity on mammalian cells have been mentioned in Tables 1 and 2. Due to their special properties, metal-based nanoparticles may exhibit toxicity under specific circumstances. They are small in size, have a large surface area, and have unique physicochemical characteristics (*Makhdoumi, Karimi & Khazaei, 2020*). These characteristics may result in interactions with biological systems that harm both the environment and human health. The size and surface area of metal nanoparticles are two important elements that affect how toxic they are (*Saifi, Khan & Godugu, 2018*). Due to their extremely small size, nanoparticles can more easily enter cells and tissues. Furthermore, studies have shown that while larger nanoparticles (NPs) enter cells *via* alternative transportation mechanisms like phagocytosis, micropinocytosis, and non-specific translocation, smaller NPs can cross cell
membranes through translocation (*Zhang, Gao & Bao, 2015*; *Zhang, Xiong & Liu, 2022*). The likelihood of interactions with biomolecules is increased by their high surface area to volume. For instance, their larger surface area makes smaller silver nanoparticles more toxic than their larger counterparts (*Recordati et al., 2015*). Many different types of nanoparticles (NPs) exist, including spheres, ellipsoids, cylinders, sheets, cubes, and rods (*Sukhanova et al., 2018*). When NPs are the same size and composition, their shape can greatly impact how they behave in terms of biodistribution, cellular uptake, deposition, and clearance (*Zare et al., 2021*). Another important element is the reactive surface of metal nanoparticles. Along with shape and size, surface chemistry also significantly impacts toxicity (*Chandran, Riviere & Monteiro-Riviere, 2017*). The surface charge influences the NP pharmacokinetics and their interactions with organelles and biomolecules, which is closely related to NP toxicity. When these nanoparticles come into contact with biological systems, their frequently highly reactive surfaces may cause the production of ROS (*An et al., 2014*). Cells and tissues may experience oxidative stress and damage due to ROS. For instance, when exposed to UV light, titanium dioxide nanoparticles (TiO$_2$ NPs) found in sunscreen lotions can produce ROS and possibly harm skin (*Musial et al., 2020*). In addition, some metal-based nanoparticles have the potential to release dangerous metal ions into the atmosphere or biological systems (*Forest et al., 2021*). Cadmium-based nanoparticles, such as cadmium selenide nanoparticles, have the potential to release toxic cadmium ions that can build up in organisms (*Chen et al., 2012*). The toxicity of nanoparticles can also be affected by the surface coatings (*Hwang et al., 2018*). Although the composition of these coatings can have an impact on toxicity, they are used to improve stability and functionality (*Ge et al., 2019*). For instance, some coatings on gold nanoparticles may lessen their toxicity while others may increase it (*Jia et al., 2017*). Another major issue is bioaccumulation. Metal nanoparticles can be ingested by living things, where they can build up in different tissues and cause toxicity (*Lopez-Chaves et al., 2018*). As an illustration, mercury-based nanoparticles, like mercury sulfide nanoparticles (HgS NPs), can build up in aquatic organisms and pose risks to human consumption and the aquatic food chain (*Ghoshdastidar & Ariya, 2019*; *Takahashi et al., 2021*). Another aspect of metal nanoparticle toxicity is their chemical reactivity with biological molecules (*Stark, 2011*). For instance, silver nanoparticles may interact with proteins and enzymes alter their activity, and result in cytotoxicity (*Flores-López, Espinoza-Gómez & Somanathan, 2019*). The solubility of metal nanoparticles can also affect how toxic they are. When in contact with moisture, poorly soluble nanoparticles like zinc oxide may release toxic zinc ions and become cytotoxic (*Pandurangan & Kim, 2015*). Figure 1 summarizes the properties accountable for nanotoxicity and the mechanisms of toxicity associated with them.

## Toxicity of carbon-based nanoparticles

Carbon is one of the components noticed earlier and is readily available. The allotropes are composed of diamond, graphite, amorphous carbon, carbon nanotubes (CNTs), graphene, and fullerenes. Due to their special physicochemical and electro-mechanical properties and biological compatibility. But they also possess harmful effects on biological processes and

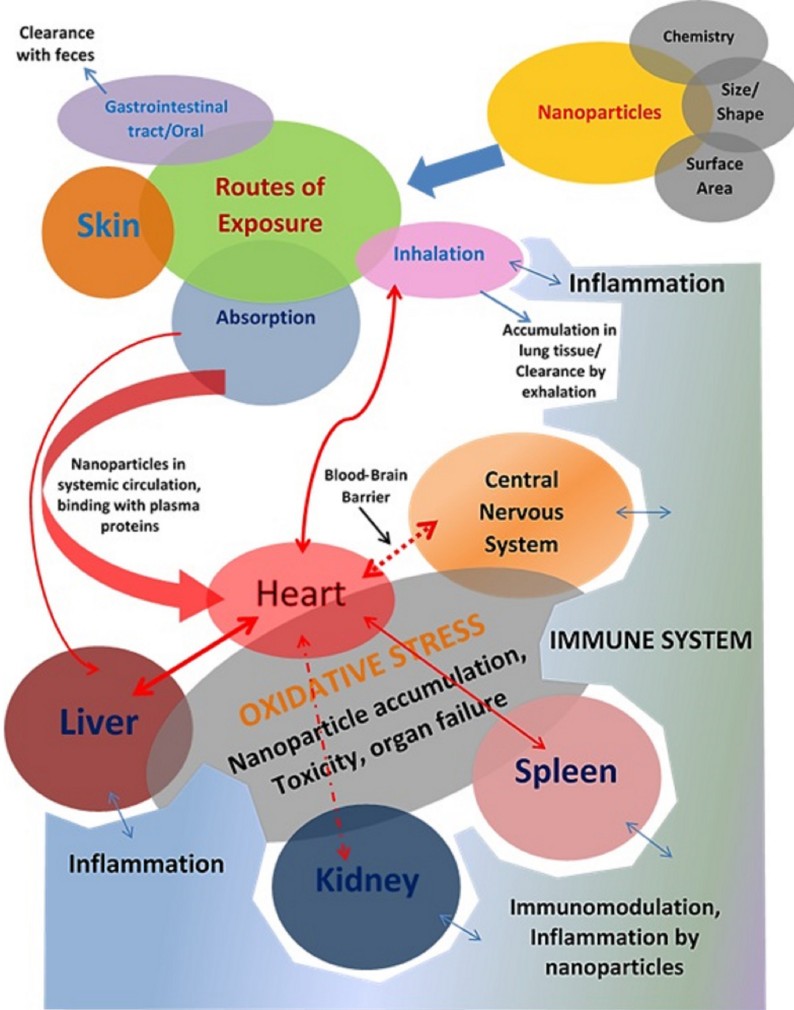

**Figure 1 Investigating the intricate and harmful relationships between nanoparticles and organ systems reveals a complicated web of possible health effects.** On the respiratory, cardiovascular, neurological, hepatic, renal, and reproductive systems, among other organ systems, nanoparticles can demonstrate a range of toxicological effects. Determining the hazards of nanoparticle exposure and creating plans to reduce unfavorable health effects require an understanding of these interactions.

cellular compartments. *Koike & Kobayashi (2006)* investigated carbon-based nanoparticles' chemical and biological oxidative effects (CBNP) and found that CBNPs with smaller sizes have greater oxidative potential than larger alveolar epithelial cells. In general, nanomaterials exist in various forms and structures such as particles, tubes, fibers, spheres, points, cubes, truncated triangles, wires, and films that influence nanoparticle (NP) kinetics and transportation in the environment (*Gonzalez-Munoz et al., 2015*; *Madannejad et al., 2019*; *Walters, Pool & Somerset, 2016*). *Srikanth et al. (2015)* assayed the cytotoxicity of four types of carbon nanomaterial (carbon nanowire, carbon nanotube, graphene, and fullerene) on L929 mouse fibroblast cancerous cells and found that graphene was the most toxic substance with an average toxicity of 52.24% followed by

CNTs, fullerene, and CNW, based on morphology, concentration, and duration of exposure. *Srikanth et al. (2015)*, data indicate that toxicity variations are related to different structural arrangements and aspect ratios. In another study, the role of surface area in oxidative and DNA damage potential of carbon-based (CB) nanomaterial was observed. They found that CB with smaller and larger surface areas demonstrate higher potencies for oxidative stress and DNA damage in rats (*Almaimani, 2021*; *Chuang et al., 2015*). Carbon nanotubes (CNTs) have drawn much scientific interest and possible applications due to their specific physical, biological, electrical, and mechanical properties. As the production and application of CNTs increase on a wide scale, the general public is more likely to be directly or indirectly exposed to CNTs, prompting substantial attention to human health and safety concerns related to CNTs. Several significant factors, such as impurities, amorphous carbon, surface charge, form, volume, aggregation, and layer numbers, explain the differences in the experimental results of nontoxicity. The *in vivo* actions and fate of CNTs can also be affected by exposure paths, including inhalation, intravenous injection, or dermal or oral exposure. Oxidative stress, inflammatory reactions, malignant transformation, DNA damage and mutation (errors in chromosome number and mitotic spindle disruption), the development of granulomas, and interstitial fibrosis are the underlying mechanisms of CNT toxicity. These results provide valuable insights into the *de novo* nature, safe application, and risk assessment of carbon nanotubes for human health (*Liu et al., 2013*).

# ROUTES OF EXPOSURE OF THE NANOPARTICLE

The human body is a complex structure of several organs, such as the heart, liver, lung, kidney, *etc*., Any damage (cellular, biochemical, or molecular level) can disrupt the entire system and lead to various pathological conditions. A drug has to penetrate the body through certain routes to cause specific harm or general toxicity (Fig. 1). These are referred to as exposure channels, which involve the skin (dermal exposure), gastrointestinal (oral exposure), and lung (inhalation exposure) exposure (*De Matteis, 2017*). Oral exposure is the most prevalent natural route of exposure to a significant range of toxicants, including nanoparticles. However, any material with a lipophilic nature is more likely to penetrate the body or systemic circulation when it comes into contact with the skin. In this scenario, inhalation is the most sensitive route of exposure, and the skin is the least sensitive. Exposure to nanomaterials in food or the atmosphere *via* the above routes. In addition to these exposure pathways, numerous clinical administration pathways for various therapies are also in operation (*Romeyke & Stummer, 2012*). These include administration using a syringe through an intravenous (IV), intraperitoneal (IP), subcutaneous (SC), intradermal (ID), transdermal (TD), intramuscular (IM), *etc*. These pathways are primarily used for therapeutic administration during treatment.

## Gastrointestinal tract

Nanoparticles are gradually making their place in the formulations of consumer goods. Food, *via* the gastrointestinal tract, is a significant source of nanoparticle exposure. Flavor

enhancers, food pigments, and supplementations can be part of the range of nanomaterials. Some non-edibles can also work as a source of oral exposure to nanomaterials. They can shed the nanomaterials used in their development. These products may include toothbrushes, baby bottles, pacifiers, containers for processed or unprocessed food items, and containers for processed or unprocessed food items (*Bergin & Witzmann, 2013*). Possible exposures to nanomaterials include consuming marine food (fish or shellfish) with accumulated nanomaterials for toxic waste ingestion or absorption (*Ahmad, Pranaw & Khare, 2019b*; *Gaiser et al., 2012*; *Toussaint et al., 2019*; *Wang et al., 2020a*). Researchers have proposed that the absorption mechanism of nanoparticles is possibly endocytosis in the gastrointestinal tract (*Frohlich & Roblegg, 2012*). In contrast to larger counterparts, nanoparticles of comparatively smaller sizes are more readily absorbed.

Silver nanoparticles, gold nanoparticles, titanium dioxide ($TiO_2$) nanoparticles, copper nanoparticles, silicon dioxide ($SiO_2$) nanoparticles, quantum dots, carbon-based nanoparticles (multi- or single-walled carbon nanotubes) and polymer/dendrimer nanoparticles (*Bergin & Witzmann, 2013*) are the materials recorded to exhibit their toxic effects after ingestion.

## Skin

Skin serves as a physical barrier for all the xenobiotics and pathogenic microorganisms. It can prevent the entry of microorganisms and hydrophilic substances. Skin remains in direct contact with the environment but interacts less with the toxic components in the ambiance. It works more as a physical barrier; it has a small surface area (1.73 $m^2$, an adult human's average body surface area). It has less blood supply and an upper layer of dead cells as a physical obstruction for most hydrophilic xenobiotics. Lipophilic substances (<600 Dalton [Da]) can easily cross the skin passively and enter systemic circulation (*Barry, 2001*; *Coates et al., 2019*). There are several reports which dealt with the skin penetration of the nanoparticles. Still, they show discrepancies in results likely related to differences in techniques and methods, laboratory conditions, and the absence of standardized evaluation protocols (*Crosera et al., 2009*; *Ghasemiyeh & Mohammadi-Samani, 2020*).

In most cases, skin exposure to nanoparticles occurs through cosmetics. Various personal care products contain nano-sized components like nano-emulsions and microscopic vesicles. The issue of dermal penetration is highlighted by applying $TiO_2$ and ZnO nanoparticles to cosmetic products. Sunscreens contain insoluble titanium dioxide ($TiO_2$) or zinc oxide (ZnO) nanoparticles, which are efficient filters of ultraviolet (UV) light (*Geppert et al., 2020*; *Lu et al., 2015*; *Mohammed et al., 2019*). Several *in vivo* toxicity studies, including *in vivo* intravenous studies, showed that $TiO_2$ and ZnO nanoparticles are non-toxic and have excellent skin tolerance (*Nohynek & Dufour, 2012*). However, this may not be the case with other nanoparticles with potential applications in cosmetics. Nohynek and Dufour, in their review, concluded that the nanoparticles in skin care products or sunscreens pose no or negligible threats to humans in terms of toxicity. Factors that may affect the absorption of any agent include (i) skin integrity and regional variation,

(ii) dimensions of orifices, aqueous pores, and lipidic fluid paths, and (iii) density of appendages (*Baroli, 2010*).

## Respiratory tract

Because of its importance for survival and exposure to a toxicant, inhalation deserves special attention. Chimney smoke, car fumes, cigarette smoke, forest fire smoke, and other harmful inhalants are inescapable in today's world. Inhalation exposure is more significant in environmental interactions. The lung and the skin both serve as interfaces between the body and its external environment, but their anatomical and physiological structures and functions are distinct. The physiology of the lungs is sensitive to even little variations in air composition. Toxic substances in the air are taken into the lungs along with the breath. Lung toxicities are very common due to the direct interaction of lung alveolar epithelium with inhaled toxicants. The sensitivity of the lung toward a variety of airborne toxic agents, including carcinogens, leads to a brisk inflammatory response (*Wong, Magun & Wood, 2016*).

The number and exposure of nanomaterials are rapidly increasing in the present environment. Most of them can cause damage to the lungs. Airborne toxicants, including nanoparticles, are significant concerns regarding human health. Many reports are available regarding the toxicological effects of nanoparticles after inhalation (*Bierkandt et al., 2018*). More specifically, several effects of inhaled nanoparticles are attributed to their (i) direct effects on the central nervous system, (ii) their translocation from the lung into the bloodstream, and (iii) their capacity to invoke inflammatory responses in the lung with subsequent systemic effects (*Borm & Kreyling, 2004*).

The nano-sized particles that can be inhaled and pose a serious threat to human health include suspended particulate matter (SPM), combustion-derived nanoparticles (CDNP), asbestos fibers, and silicon nanoparticles (*Borm et al., 2006*). CDNPs are generated in some scenarios, including internal combustion engines, large-scale coal burning for power generation, and industrial processes where they often could be produced along with larger particles. CDNPs may include diesel exhaust particulates, welding fumes, nanoparticulated carbon black, and coal fly ash. These all cause oxidative stress and lung inflammation after inhalation (*Diabaté et al., 2011*). Asbestosis results from inhalation exposure to nano-sized asbestos fibers, resulting in mesothelioma (cancer of the pleural lining of the lung), lung cancer, and laryngeal cancer (*Offermans et al., 2014*; *Ross, 2014*). Most lung ailments are associated with acute or chronic inflammation. The main reason behind the sensitivity of the lung lies in its anatomy, which is constituted by a large number of alveoli, increasing its surface area up to 75 m$^2$, and a large number of vasculatures. High blood flow in the lungs makes it more vulnerable to inflammatory responses and the primary organ responsible for most nanoparticle-related toxicities. Chronic obstructive pulmonary disease (COPD), an accumulation of emphysema, bronchitis, and fibrosis, is primarily a chronic inflammatory disease. Other examples of lung inflammatory ailments include bronchiolitis, asthma, acute inflammation after toxic exposure, pneumonitis, *etc*. Most of

the lung cancer cases are also associated with chronic lung inflammation-related episodes. It has been found that COPD and lung cancer positively correlate in most cases. Such an episode's exposure to any nano-sized particle may impose similar or more serious consequences. Several studies have indicated that nanoparticles react aggressively when exposed to biological systems due to their physic-chemical properties being different from their normal forms (*Cheng et al., 2013*).

## BIODISTRIBUTION

Due to their physicochemical properties, which are different from their crude counterparts, the nanomaterials may have a different distribution pattern in a biological system. Several pharmacokinetics studies provide a set of their distribution pattern, and the information is efficiently applied in the nanomaterials' medical diagnostics, pharmacology, and toxicology (*Madru et al., 2013*; *Singh & Pai, 2014*). Studies indicate that the distribution of the nanoparticles initially depends on the particle shape and size (*Kreyling et al., 2014*; *Toy et al., 2014*). The dependence of the biodistribution of nanoparticles on their physical properties signifies the role of the physicochemical nature of the nanomaterials in toxicological effects and the different toxicological nature of the nanomaterials when compared with their crude forms (*Duan & Li, 2013*). Reports also indicate the dependency of the biodistribution of nanoparticles on phagocytosis. In a rat model, *Li et al. (2014)* showed that the distribution of polyethylene glycol-coated polyacrylamide nanoparticles across highly perfused organs depends on phagocytosis. In their study, various target organs include the spleen, liver, bone marrow, lungs, heart, and kidneys. However, apart from phagocytosis, the nanoparticles also interact with the plasma proteins as soon as they enter the physiological environment. According to their specific functions, adsorbed proteins can be divided into opsonin and dysopsonins. Opsonin often induces the rapid blood clearance of nanoparticles, while dysopsonins benefit prolonged blood circulation (*Gao & He, 2014*). Nanoparticle-protein interactions can greatly influence biodistribution (*Baimanov, Cai & Chen, 2019*; *Ovais et al., 2020*). Though biological tools like phagocytosis and plasma proteins contribute to the pharmacokinetics of nanoparticles, the physicochemical properties of the nanoparticles themselves and the way how they are modified for delivery into the system also directly influence their distribution in the organ system (*Ahmad et al., 2021*; *Patra et al., 2018*). In certain cases, polyethylene glycol (PEG) density and lipids on the nanoparticle surface, such as lipid-calcium-phosphate (LCP) nanoparticles, can influence the pharmacokinetics and biodistribution of the nanoparticle. *Liu, Hu & Huang (2014b)* reported that delivering LCP nanoparticles to hepatocytes depends on the concentration of PEG and the surface lipids. LCP nanoparticles could be directed from hepatocytes to Kupffer cells by decreasing PEG concentration on the particle surface. This study signifies the role of the physical characteristics of the nanoparticles in their pharmacokinetics and biodistribution. It boils down to that a complex array of interrelated, physicochemical, and biological factors of the nanoparticles control the biodistribution of nanoparticles. These factors should be considered while planning a pharmacokinetics study of any nano-sized material. A

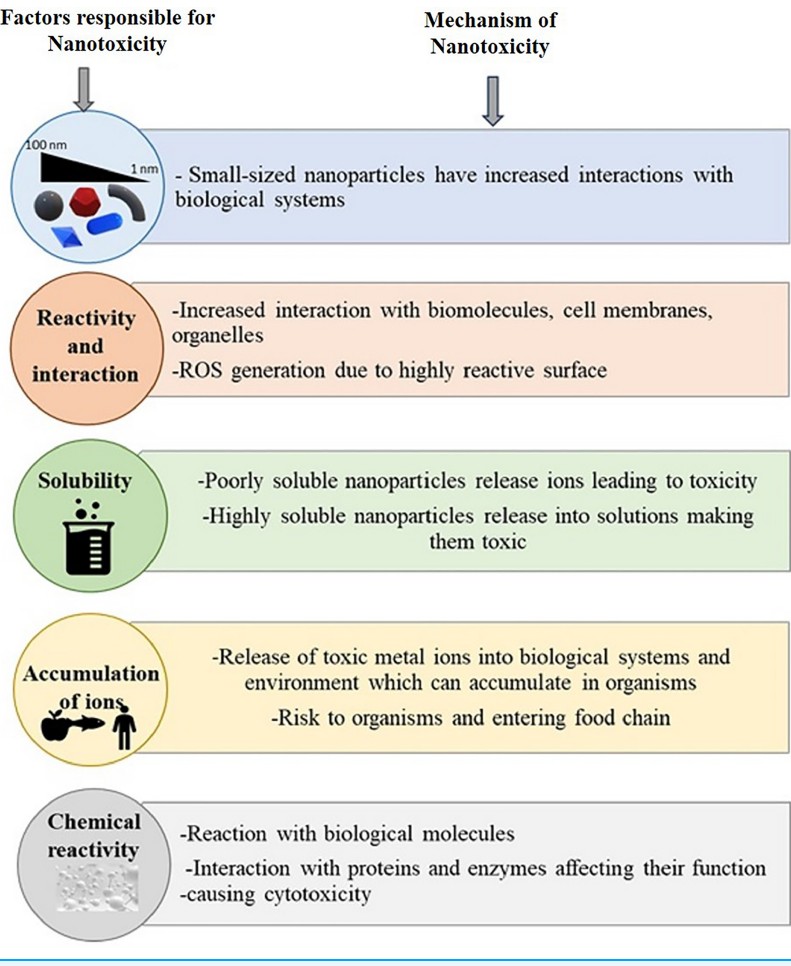

**Figure 2 Summary of properties responsible for nanotoxicity and mechanisms involved.** Size. interaction, solubility, accumulation, and chemical reactivity of nanoparticles play crucial roles in nanotoxicity.

schematic representation of the factors and mechanism of nanoparticle toxicity is given in Fig. 2.

# MECHANISMS OF TOXICITY OF NANOPARTICLES

Nanoparticle causes adverse health effects by different means. Some possible toxicity mechanisms include oxidative stress, inflammation, genetic damage, and immune system dysfunction (*Crisponi et al., 2017*; *Manke, Wang & Rojanasakul, 2013*). Lysosomes often target NP toxicity because most pass into the cells by endocytosis. NPs produce lysosomal dysfunction by cytoskeleton disruption, alkalization of the lysosomal lumen, or NP overload (*Crisponi et al., 2017*; *Stern, Adiseshaiah & Crist, 2012*).

## Genotoxicity

Like any other general chemical forms, nanomaterials have the very same potential to modulate immune responses in an organism and the potential to alter the constitution of genetic material. Several researchers have proposed disturbances in nucleic acid sequences

or simple damage to the DNA strands by nanoparticles (*Akhtar et al., 2016*; *Ghosh, Ahmad & Khare, 2018*; *Ghosh, Ahmad & Khare, 2019*; *Jeevanandam et al., 2018*; *Kumar & Dhawan, 2013*). The main mechanism of genotoxicity by nanomaterials in *in-vitro* or *in-vivo* studies is oxidative damage, which is proposed to be the main mechanism of most nanomaterials for their toxic potential (*Akhtar et al., 2016*; *Dobrzynska et al., 2014*; *Kumar & Dhawan, 2013*; *Powell et al., 2010*; *Xia, Li & Nel, 2009*). Reactive oxygen species and other free radicals/oxidative species are very well known to cause DNA damage (*Juan et al., 2021*; *Kunwar & Priyadarsini, 2011*; *Shi & Dansen, 2020*). Other mechanisms may include the direct interaction of nanomaterials with DNA/RNA and the release of toxic ions. A range of nanoparticles is reported or suggested to exhibit genotoxic potentials in *in-vitro, in-vivo*, or *in-silico* studies (*Kumar & Dhawan, 2013*). The reported nanoparticles for their genotoxic potential include but are not limited to, nano-sized silver, titanium dioxide, nickel, copper oxide, carbon nanotubes, *etc.*

## Immunomodulation

Activation of neutrophils is considered the central event in acute inflammation episodes (*Dwivedi et al., 2011*; *Rahman et al., 2011*; *Rahman et al., 2008*). *Goncalves, Chiasson & Girard (2010)* reported the *in-vitro* activation of the human neutrophils by titanium dioxide (TiO$_2$) nanoparticles. The same study also demonstrated that TiO$_2$ markedly and rapidly induced tyrosine phosphorylation events, including phosphorylation of two key enzymes significantly. The nanoparticles exhibit their immune-modulatory properties. Because of their small size, nanoparticles can penetrate and accumulate much deeper cellular and even sub-cellular boundaries, which can cause many undesirable and uncharacterized effects. Nanoparticles may bind to serum proteins and act as haptens to induce an immunological response, including activation of the complement system and disturbance in the balance of the helper T cells. They may escape the phagocytic activity of macrophages. They may discompose the adaptive or innate immunity leading to a hyper-activated immune state or even a suppressed immune phenotype, p38 mitogen-activated protein kinase (MAPK) and extracellular signal-regulated kinases-1/2 (Erk-1/2) (*Goncalves, Chiasson & Girard, 2010*). However, this *in-vitro* study exhibits the influence of the TiO$_2$ nanoparticles on the molecular pathways involved in regulating inflammation and other immunological events. The probabilities of such incidences can be extrapolated to the *in-vivo* conditions. Different types of materials may have different effects on the immune system. For instance, cobalt and nickel nanoparticles have inflammatory effects, whereas hydroxyapatite crystals release TNF-α from macrophages, activating other phagocytes (*Dwivedi et al., 2009*). A hyperactive immune state or even a suppressed immunophenotype may result from nanoparticles that evade the phagocytic activity of macrophages, bind to serum proteins and act as haptens, activate complement cascades, upset the Th1/Th2 balance, cause hemolysis or thrombogenicity, or disrupt adaptive or innate immunity (*Dwivedi et al., 2011*). Certain NPs build up in local lymph nodes, where dendritic cells can take them up and process them. They then interact with self-proteins to change their antigenicity, which in turn causes altered immunological reactions, including autoimmunity (*Di Gioacchino et al., 2011*; *Dwivedi et al., 2009*).

Studies conducted *in vitro* showed that NPs could direct the production of cytokines toward either Th1 (Pl, Pd, Ni, Co) or Th2 (Ti, mw, and sw Carbon) production patterns (*Di Gioacchino et al., 2011*). *Delogu et al. (2012)* investigated the impact of functionalized CNTs on immune cells and found strong activation of NK cells. Thus, considerable data suggest that NPs exert immunotoxicity by affecting different immune system effector cells.

## Oxidative stress

Oxidative stress is caused either by an increase in the production of reactive oxygen species (ROS) depletion in the ability of cells to destroy ROS, or a combination of both. Several *in vivo* and *in vitro* studies have pointed out that ROS generation and oxidative stress are essential in NP-induced toxicity (*Manke, Wang & Rojanasakul, 2013*; *Oberdörster, Stone & Donaldson, 2007*; *Shvedova et al., 2012*). *Federici, Shaw & Handy (2007)*, found increased ROS generation in rainbow trout by exposure to TiO$_2$ NPs in a concentration-dependent manner. Nanoparticles induce oxidative stress in several ways (*Crisponi et al., 2017*). It can be generated directly from the surface of nanoparticles (*Huang, Cambre & Lee, 2017*). Transition metal (iron, copper, chromium, vanadium, *etc.*) nanoparticles can generate reactive oxygen, acting as catalysts in Fenton-type reactions (*Risom, Møller & Loft, 2005*). Nanoparticles may enter mitochondria (*Crisponi et al., 2017*; *Nguyen et al., 2015*) and alter their function, producing ROS. Activation of inflammatory cells, such as alveolar macrophages and neutrophils, induced by phagocytosis of nanoparticles, can lead to the generation of reactive oxygen species and reactive nitrogen species (*Völs et al., 2022*). Oxidative stress corresponds with the physicochemical reactivity of NP. It involves mitochondrial dysfunction, depletion of antioxidant enzymes, lipid peroxidation of cellular membranes, protein modification, and DNA damage associated with cell and tissue injury (*Samrot & Noel Richard Prakash, 2023*; *Pacheco & Buzea, n.d.*; *Crisponi et al., 2017*). NP-driven ROS generation also contributes to the activation of cell signaling pathways, inflammatory cytokine and chemokine expressions, and specific transcription factor activation, which may lead to pathological consequences (*Crisponi et al., 2017*; *Manke, Wang & Rojanasakul, 2013*; *Medina et al., 2007*). A schematic representation of the toxic effects of nanoparticles at the cellular level is given in Fig. 3.

## Inflammation

ROS and inflammation demonstrate an interdependent relationship in the case of exposure to NP (*Manke, Wang & Rojanasakul, 2013*). Inflammation is the typical response of the body to injury. When generated in moderation, inflammation helps fight against infection and eliminate foreign invaders; however, in excess and chronic conditions, it can lead to pathological conditions leading to the cause of many diseases (*Jeong et al., 2022*). Various *in vitro* and *in vivo* experiments suggest that exposure to nanoparticles is associated with mild to severe forms of inflammation, depending upon the size and composition of the nanoparticle (*Aldayel et al., 2023*; *Zhou, Jin & Ma, 2023*).

An intricate cascade of intracellular and extracellular processes regulates inflammation. Pro-inflammatory mediators, also known as cytokines, are secreted in response to oxidative stress and are largely responsible for the immune system's activation

**Figure 3 Toxicity of nanoparticles at the cellular level.** A human is exposed to these nanoparticles by different means, which enters into the cells by lysosomes mediated endocytosis, which results in lysosome dysfunction and ROS generation. These ROS inside the cells causes lipid peroxidation of the membrane, protein modification, mitochondrial dysfunction, and DNA damage.

(*Wolf-Grosse et al., 2017*; *Rahman et al., 2015*). Inflammation has also been demonstrated to directly cause toxicity and increase cell death through the development of toxic by-products of inflammation such as ROS and complement proteins and through receptor-induced necrosis (*Khanna et al., 2015*; *Wallach, Kang & Kovalenko, 2014*). Increased production of pro-inflammatory proteins IL-6, IL-8, and MCP-1, as well as activation of the JNK and p53 pathways, have been linked to silica nanoparticle toxicity (*Liu & Sun, 2010*). The PI3-K/Akt/mTOR signaling cascade is vital for cell survival and proliferation because it controls the cell cycle. PI3-K signaling was reported to promote the upregulation of Cox-2, iNOS, and pro-inflammatory cytokines (IL-6, IFN-γ, TNF-α, IL-17, and regulatory cytokine IL-10) in macrophages with exposure to zinc oxide nanoparticles (*Khanna et al., 2015*; *Roy et al., 2014*). As a result, there is evidence that NPs can have an inflammatory effect by activating many signaling pathways and producing reactive oxygen species.

## METHODS FOR ASSESSING THE TOXICITY OF NANOMATERIALS

To assure a responsible and sustainable growth of nanotechnology, the health, and safety associated issues of engineered nanomaterials and related products need to be addressed at a rate commensurate with nanotechnology's expansion. The toxicity assessment of any nano-sized material follows the same path as any other chemical. Both *in-vitro* and *in-vivo* approaches are considered while assessing the safety of a nano material-related product

**Table 3 Aspects of assessment of characterization of nanomaterials.**

| S. no. | Aspect of assessment | Description | References |
|---|---|---|---|
| 1. | Nanomaterial characterization | Characterization of Nanomaterials describing the size, shape, surface area, surface chemistry, composition, stability, and other physical and chemical characteristics of nanomaterials. This facilitates understanding of the possible relationships with biological systems. | *Jayawardena et al. (2021)*, *Banerjee et al. (2016)* |
| 2. | *In vitro* studies | Testing the toxicity of nanomaterials through cell culture experiments. This involves evaluating different cell types' inflammatory responses, genotoxicity, oxidative stress, proliferation, apoptosis, and viability. | *Ganguly, Breen & Pillai (2018)*, *Verma et al. (2021)*. |
| 3. | *In vivo* studies | Animal studies evaluate nanomaterials' pharmacokinetics, biodistribution, and toxicity after exposure *via* various routes (*e.g.,* ingestion, injection, or inhalation). This aids in comprehending long-term toxicity and systemic effects. | *Lopez-Chaves et al. (2018)*, *De Matteis (2017)*. |
| 4. | Genotoxicity assessment | Assessing the effects of immune system exposure to nanomaterials, including cytokine production, immune cell activation, and hypersensitivity reactions. To evaluate the possible immunomodulatory effects of nanomaterials is essential. | *Landsiedel et al. (2022)*, *Siivola et al. (2022)*, *Elespuru et al. (2018)*. |
| 5. | Neurotoxicity assessment | Investigating how exposure to nanomaterials affects the nervous system, including how it affects behavior, neuronal viability, morphology, and neurotransmitter release. This is critical for assessing the possible neurotoxic impact of nanoparticles. | *Boyes & van Thriel (2020)*, *Sofranko et al. (2021)*, *Liu et al. (2020)* |
| 6. | Environmental fate and transport | Examining the aggregation, sedimentation, bioaccumulation, and possible ecological effects of nanomaterials in environmental matrices like soil, water, and air. This aids in determining exposure pathways and environmental risks. | *Suhendra et al. (2020)*, *Baalousha et al. (2016)*, *Rawat et al. (2018)*. |
| 7. | Risk assessment and management | Integrating data from toxicological research to evaluate possible risks of exposure to nanomaterials and create risk-reduction plans. This covers exposure assessment, dose-response modeling, and regulatory compliance. | *Trump et al. (2018)*, *Laux et al. (2018)*. |
| 8. | Nanomaterial interactions with biomolecules | Investigation of how nanomaterials interact with biomolecules such as proteins, lipids, and nucleic acids, which can affect their biological behavior and toxicity. This helps in gaining an understanding of their mode of action and potential adverse effects. | *Nienhaus, Wang & Nienhaus (2020)*, *Fadeel (2019)*, *Wang, Cai & Chen (2019)*. |
| 9. | Long-term exposure studies | Analyzing the interactions between biomolecules, including proteins, lipids, nucleic acids, and nanomaterials, can influence these materials' toxicity and biological activity. This aids in comprehending their mechanism of action and possible side effects. | *Abarca-Cabrera, Fraga-García & Berensmeier (2021)*, *Auría-Soro et al. (2019)*, *Casalini et al. (2019)*. |
| 10. | Standardization and guidelines | Establishing standardized protocols and guidelines for evaluating nanotoxicology to guarantee the coherence and comparability of study findings. This makes evaluating risk and making regulatory decisions about nanomaterials easier. | *Rasmussen et al. (2019)*, *Fernández-Cruz et al. (2018)*, *Potthoff et al. (2015)*. |

(Table 3). The related safety concerns are also increasing with the emerging applications of nanotechnology in industries and household products. Human health concerns for nanomaterials have been established historically by epidemiological and clinical studies on naturally occurring fibers and particles such as asbestos and silica (*Ahmad, Mishra & Sardar, 2014b*; *Alam et al., 2018*; *Ghosh, Ahmad & Khare, 2019*; *Hillegass et al., 2010*; *Mohajerani et al., 2019*). The nano-sized fibers of asbestos and silica enter the body through inhalation and cause a range of ailments, including lung fibrosis and
cancer. Table 3 depicts the various aspects of assessment of the characterization of nanomaterials.

## *In vitro* methods

The *in vitro* model, which mainly includes cultured cells, provides a rapid and effective assessment of some toxicological endpoints associated with nanomaterial exposure (*Eskes et al., 2017*; *Natoli et al., 2012*). *In-vitro* assessments allow mechanism-based toxicity evaluations and provide precise information on how nanoparticles interact with biological systems in many ways (*Gupta & Xie, 2018*; *Savage, Hilt & Dziubla, 2019*). *In-vitro* assays include (i) cell viability/cytotoxicity assay, mechanistic estimations, (ii) microscopic analyses, (iii) gene expression evaluation, and (iv) genotoxicity estimations *etc.*

Several assays have been mentioned to assess the effects of nanomaterials on cell viability or cytotoxicity. The review of the *in-vitro* studies by *Hillegass et al. (2010)* recommends collecting standard growth curve data to determine the baseline growth properties of selected cells. Various *in-vitro* estimations are performed to estimate a chemical or agent (Including nanomaterials). The assays include (i) trypan blue exclusion assay, (ii) microculture tetrazolium assay (MTA), (iii) clonogenic assay or colony-forming efficiency (CFE), (iv) lactate dehydrogenase (LDH) Assay, (v) TdTdUTP nick end labeling (TUNEL) and apostate assays. However, several other assays can also be employed to estimate cytotoxicity (*Balouiri, Sadiki & Ibnsouda, 2016*). Other *in vitro* nanotoxicity assays include the examination of lipid peroxidation to elucidate the role played by oxidative stress and methods to investigate apoptosis, including cytochrome c release from mitochondria and caspase activation (*Hillegass et al., 2010*).

*In-vitro* assays can also be exploited to estimate the nanomaterials' cytotoxicity and their probable effect on cell proliferation (*Leudjo Taka et al., 2021*). For cellular proliferation estimation, the assays that have been recommended include (i) DNA content estimation, (ii) Incorporation of Bromodeoxyuridine (BrdU), (iii) Ki-67 nuclear antigen estimation, (iv) proliferating cell nuclear antigen (PCNA).

The genotoxicity of the nanomaterials can be evaluated in both *in-vitro* and *in-vivo* systems. Estimations in *in-vitro* are much simpler and easier when compared with the *in-vivo* systems. *In-vitro* systems are less time-consuming, and one can get the results in a short period (*Barabadi et al., 2019*; *Elespuru et al., 2018*). The *in-vitro* assays for genotoxicity include (i) Ames assay in *Salmonella typhimurium*, and *Escherichia coli*, (*Pan, 2021*) (ii) oxidized guanine bases estimation, (*Cui et al., 2013*) (iii) analysis of chromosomal aberration, (*Clare, 2012*) (iv) micronuclei assay (*Doherty, 2012*) (v) single cell gel electrophoresis (comet) assay (*Karlsson, 2010*).

## *In-vivo* methods

Over the last decade, nanotoxicology methods have mostly relied on *in vitro* cell-based estimations, which provide a good amount of information related to mechanistic approaches. Still, they do not account for the complexity of *in vivo* systems concerning biodistribution, metabolism, hematology, immunology, target organs, and neurological consequences. Compared to many reports published about nanoparticles, only 3% are

devoted to studying their critical biological effects *in vivo* (*Greish, Thiagarajan & Ghandehari, 2012*; *Hussain et al., 2020*).

Various nanotoxicological studies incorporate the traditional *in-vivo* models for safety evaluation. The models include zebrafish, Xenopus embryos, mice, rats, and sometimes a primate (Macaque) (*Yong et al., 2013*).

Zebrafish (*Danio rerio*) is extensively used as an animal model for toxicological studies (*Hill et al., 2005*; *Sipes, Padilla & Knudsen, 2011*). The Zebrafish genome project has placed zebrafish in an attractive position for use as a toxicological model. The zebrafish embryo is also a useful small model for investigating vertebrate development because of its transparency, low cost, transgenic and morpholino capabilities, conservation of cell signaling, and similarities with mammalian developmental phenotypes (*Sipes, Padilla & Knudsen, 2011*). The zebrafish has also been proposed and used as an animal model in toxicological studies associated with nano-sized materials (*Bohnsack et al., 2012*; *Fako & Furgeson, 2009*).

Mouse and rat models have been used in life science research for several decades. They are also used for toxicological evaluations of industrial and environmental chemicals. The main reason for the selection of mice as a model is its genomic similarity with humans, and it makes the primary base for the wide use of mice in the life sciences, pharmacological and toxicological research; the mouse as a mammalian model provides corresponding experimental conditions and analogous results to humans, though with certain limitations. The use of animal models sometimes raises certain ethical issues. However, it has been proven a successful animal model for preclinical research. Nanotoxicological studies have also adopted the mouse as a model for estimating the toxic potential of nanomaterials (*Bahadar et al., 2016*; *Ferreira, Cemlyn-Jones & Cordeiro, 2013*; *Lam et al., 2004*). Rats are also used for experimental purposes similarly (*Devoy et al., 2020*; *Ferreira, Cemlyn-Jones & Cordeiro, 2013*; *Konoeda et al., 2020*; *Mendonca et al., 2016*; *Warheit, 2004*). Animal models are required to estimate the absorption of material into the biological system, its distribution in the body, its fate after metabolism, and, ultimately, its modes of elimination from the body (ADME studies). Absorption, distribution, metabolism, excretion, pharmacokinetics (ADME/PK), and carcinogenic and teratogenic studies are also required to evaluate nanomaterial toxicity (*Greish, Thiagarajan & Ghandehari, 2012*; *Raja et al., 2017*; *Zielinska et al., 2020*). While studying the toxicological effects of nanoparticles, one should cautiously consider the physicochemical properties that include size, shape, surface area, surface charge, charge density, chemical composition, the density of structure, presence of pores, and surface activating sites. These characteristics of nanomaterials may influence their toxicological effects.

## CONCLUSION AND FUTURE PROSPECTS

In conclusion, worries regarding nanomaterials' possible negative impacts on the environment and human health have grown as a result of their rapid development across a variety of industries. To allay these worries, the safe development and use of nanomaterials must come first. To ensure the proper use of nanomaterials and to promote their creation, regulatory rules must be established and reinforced. Ensuring the safety of nanomaterials,

especially those destined for medical and diagnostic applications, requires meticulous evaluation of their physicochemical properties during the fabrication process. This thorough characterization helps identify possible hazards and provides information for the creation of safer nanomaterials. Further investigation into the appropriate handling, use, and storage of recently developed nanomaterials is essential to reduce exposure in the workplace and avoid accidental release into the environment. Toxicologists are actively looking into the dangers and hazards that come with nanomaterials, and developing sophisticated toxicological methodologies is crucial to a quicker and more precise assessment. The toxicity of nanomaterials is greatly influenced by their physicochemical characteristics, including size, chemical composition, surface shape, and aggregation state. This emphasizes the need to develop environmentally benign, non-toxic nanoparticles. Furthermore, targeted research efforts, the application of systems biology methodologies, and *in-silico* techniques are needed to comprehend the intricate relationships between these features and nanoparticle toxicity. These cutting-edge techniques can aid in the creation of stricter safety laws and safer nanomaterials by offering insightful information about anticipating and reducing potential risks related to nanoparticles. In summary, companies, regulatory agencies, and scientists must work together to protect the environment and public health as nanotechnology advances. Responsible nanomaterial design, in-depth characterization, and creative research are key components of our approach to ensuring the safe and sustainable advancement of nanotechnology for the good of society. In summary, companies, regulatory agencies, and scientists must work together to protect the environment and public health as nanotechnology advances. Responsible nanomaterial design, in-depth characterization, and creative research are key components of our approach to ensuring the safe and sustainable advancement of nanotechnology for the good of society. Understanding the interdisciplinary nature of nanotoxicology research and its dynamic character is crucial. To thoroughly evaluate the safety of nanomaterials, nanotoxicology incorporates concepts from several scientific disciplines, including toxicology, materials science, chemistry, biology, and environmental science. This assessment also recognizes that nanotechnology is a dynamic field, with quick developments producing new nanomaterials with a wide range of uses. When novel nanomaterials are created and released into the market, it is critical to methodically and proactively assess any possible dangers. Furthermore, this review emphasizes how crucial it is to take into account nanomaterials' possible indirect effects on ecosystems and public health in addition to their direct toxicity. This entails evaluating nanomaterials' environmental fate and transit, environmental persistence, and potential for bioaccumulation in food chains. The regulatory elements of nanotoxicology are also covered in this review, emphasizing the necessity of strong risk assessment frameworks and regulatory rules to guarantee the safe use of nanomaterials. Global regulatory bodies are becoming more aware of the special difficulties that nanoparticles provide and are attempting to develop uniform guidelines for handling and evaluating them. The introduction concludes by highlighting the contribution that cooperation and knowledge exchange make to the advancement of nanotoxicology research. Together, scientists, engineers, regulators, and politicians can tackle the multifaceted issues surrounding the

safety of nanomaterials and advance responsible nanotechnology innovation by cultivating interdisciplinary cooperation. In conclusion, this review offers a thorough review of nanotoxicology, addressing important ideas, difficulties, and future directions in the discipline. The review article establishes the framework for a thorough investigation of nanomaterial toxicity and its consequences for human health and the environment by recognizing the multidisciplinary nature of nanotoxicology research, the ever-changing field of nanotechnology, and the significance of regulatory oversight.

### Funding
The Deanship of Scientific Research at Umm Al-Qura University supported this work by Grant Code: (23UQU4330890DSR002). The funders had no role in study design, data collection and analysis, decision to publish, or preparation of the manuscript.

### Grant Disclosures
The following grant information was disclosed by the authors:
Deanship of Scientific Research at Umm Al-Qura University: 23UQU4330890DSR002.

### Competing Interests
The authors declare that they have no competing interests.

### Author Contributions
- Wajhul Qamar conceived and designed the experiments, performed the experiments, analyzed the data, prepared figures and/or tables, authored or reviewed drafts of the article, and approved the final draft.
- Shweta Gulia conceived and designed the experiments, prepared figures and/or tables, authored or reviewed drafts of the article, and approved the final draft.
- Mohammad Athar conceived and designed the experiments, authored or reviewed drafts of the article, and approved the final draft.
- Razi Ahmad conceived and designed the experiments, performed the experiments, analyzed the data, prepared figures and/or tables, authored or reviewed drafts of the article, and approved the final draft.
- Mohammad Tarique Imam conceived and designed the experiments, authored or reviewed drafts of the article, and approved the final draft.
- Prakash Chandra conceived and designed the experiments, authored or reviewed drafts of the article, and approved the final draft.
- Bhupendra Pratap Singh conceived and designed the experiments, authored or reviewed drafts of the article, and approved the final draft.
- Rizwanul Haque conceived and designed the experiments, authored or reviewed drafts of the article, and approved the final draft.
- Md. Imtaiyaz Hassan conceived and designed the experiments, authored or reviewed drafts of the article, and approved the final draft.

- Shakilur Rahman conceived and designed the experiments, performed the experiments, analyzed the data, prepared figures and/or tables, authored or reviewed drafts of the article, and approved the final draft.

## Data Availability

This is a literature review.

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
