# Peer review of "An insight into impact of nanomaterials toxicity on human health"

_PeerJ, doi:10.7717/peerj.17807_

## Round 0.1 · original submission · Major Revisions

Dear authors, thank you for your submission. Please, refer to the reviewers' comments and attachments for further details.

**Language Note:** PeerJ staff have identified that the English language needs to be improved. When you prepare your next revision, please either (i) have a colleague who is proficient in English and familiar with the subject matter review your manuscript, or (ii) contact a professional editing service to review your manuscript. PeerJ can provide language editing services - you can contact us at [email protected] for pricing (be sure to provide your manuscript number and title). – PeerJ Staff

·

Basic reporting

Is the review of broad and cross-disciplinary interest and within the scope of the journal?
Yes. The article discusses several physicochemical aspects of nanomaterials, their potential to interact with and harm humans, and how they can cause damage.

Has the field been reviewed recently? If so, is there a good reason for this review (different point of view, accessible to a different audience, etc.)?
Nanotechnology is developing and its demand is continuously increasing. Therefore, new data and questions are raised daily, supporting the need for review papers like this.

Does the Introduction adequately introduce the subject and make it clear who the audience is/what the motivation is?
Yes.

Experimental design

Is the Survey Methodology consistent with a comprehensive, unbiased coverage of the subject? If not, what is missing?
Yes.

Are sources adequately cited? Quoted or paraphrased as appropriate?
Yes. There are no issues.

Is the review organized logically into coherent paragraphs/subsections?
Partially. In my opinion, some reordering could enhance the quality of the paper and make it clearer for readers. I suggest that on the attachment.

Validity of the findings

Is there a well developed and supported argument that meets the goals set out in the Introduction?
Yes. The article could support researchers and decision-makers in understanding how nanoparticles could threaten human health.

Does the Conclusion identify unresolved questions / gaps / future directions?
Partially. The authors conclude and close the article, but I missed some statements that reinforce the need to assure the safe use of nanoparticles despite the known advantages of its use.

Additional comments

The article is presented in good condition, with proper English language.

It is possible to see a vast quantity of references read by the authors to write the submitted manuscript. Most of the references are recent and is possible to see an effort of the authors in searching updated articles. However, old references are still being used and could be replaced by newer ones.

Short sentences must be used in order to make information clear.

In the entire article, the authors should only use the toxicity on non-human species to clearly support or relate the effects on human health - the main subject of the text. If this is not the case, avoid using the information.

Some words and expressions must be standardized, e.g. 'physicochemical' without the hyphen, 'in vivo' and 'in vitro' without the hyphen and in italic form.

This paper has great potential and could be improved upon before Acceptance.

Detailed suggestions can be found in the attachments.

Reviewer 2 ·

Basic reporting

The MS 'An insight into impact of nanomaterials toxicity on human health'
attempts to fill the gaps in concepts and understanding of nanotoxicology by shedding light on the possible health risks of nanomaterials and to provide up to date information on the field.

Below are my comments:
The MS text is clear and mostly unambiguous. But, instead of using long sentences the author should try shorter ones wherever possible to make things easier to understand for the reader. The English language is fine although there are some grammatical mistakes that should be corrected.
The references should be more up-to-date and the focus should be more on the literature references from the recent years.
The MS has too few figures and Tables for a review article.
The review meets the journal scope. Unfortunately, the review is not sufficiently broad and cross disciplinary in nature and lacks in depth information and more recent references. The review in the current state is not publishable. Moreover, this field has been reviewed recently:
1. M. Roberto, M., & A. Christofoletti, C. (2020). How to Assess Nanomaterial Toxicity? An Environmental and Human Health Approach. IntechOpen. doi: 10.5772/intechopen.88970
2. Xuan, L, Ju, Z, Skonieczna, M, Zhou, P-K, Huang, R. Nanoparticles-induced potential toxicity on human health: Applications, toxicity mechanisms, and evaluation models. MedComm. 2023; 4:e327. https://doi.org/10.1002/mco2.327
The introduction is acceptable and motivation clear.
Key words are similar and the authors are encouraged to find a better set of key words.
An abbreviation should be introduced in its full form at the first occurrence in the MS i.e. NPs

Experimental design

A more rigorous investigation is needed and the survey methodology should be more comprehensive with sufficient coverage of the field from literature references in the recent years, say past five years. Although the organization of the MS and sub-headings are good there is not enough discussion and mostly superficial information. Unfortunately, the MS does not full-fill the goals set out in the rationale. The authors should give more detailed information that is supported by sufficient figures, tables and critical analysis of the data collected.

Validity of the findings

The MS has a very poor Conclusions section. A more detailed presentation and discussion that supports the goals set out in the introduction is required. Unresolved gaps and questions should be clearly identified by the authors with suggestions and remedies for future directions in this field.

---

## Round 0.2 · Major Revisions

Dear authors,

Many thanks for the improvement in your manuscript. However, I do also believe that it still requires some more effort and creativity in order to add to the existing literature. Additionally, please notice that I cannot accept a manuscript until all the language issues are resolved! Please, refer to the reviewers' comments for further details.

**Language Note:** The Academic Editor has identified that the English language must be improved. PeerJ can provide language editing services - please contact us at [email protected] for pricing (be sure to provide your manuscript number and title). Alternatively, you should make your own arrangements to improve the language quality and provide details in your response letter. – PeerJ Staff

·

Basic reporting

The review is interesting and meets with the scope of the journal.

There are other reviews on the same subject, however, as nanotechnology advances quickly, nanotoxicology also does. So, the manuscript is useful and will have a great audience.

The introduction is adequate to the present proposal.

Experimental design

The study design was enhanced by the authors, bringing newer references. The article's organization was already good.

Validity of the findings

Yes. The manuscript meets these demands.

Additional comments

The insertion of other figures and tables makes the article more substantial. However, figure 1 has another publication as its source, so it is necessary to check authorization or copyright issues.

The article was improved and meets the requirements for publication in PeerJ.

Reviewer 2 ·

Basic reporting

Although the authors have now used shorter sentences there are quite a few grammatical mistakes that need to be corrected.
The authors should increase the number of Tables; nanomaterials toxicity assessment is very important and a table on this topic with recent references would be very helpful and enhance the MS. Tables 1 and 2 should have more updated references with the most recent information.
The legends for Fig.2 and 3 should be more descriptive.
It is good to note that the authors have included more references from the recent years in the revised MS. But, unfortunately the text of the MS has not improved and the review lacks in depth information and comprehensive discussions. The review in the current state is not publishable.

Experimental design

After carefully reading the revised version I find that the authors have made no improvement in the MS text. The words are only rearranged! A rigorous investigation and analysis of the data from recent literature is needed. More comprehensive and detailed discussions are required under all headings. The MS in its current form does not provide any new insights and the MS does not full-fill the goals set out in the rationale. The authors should give more detailed information that is supported by sufficient figures, tables and critical analysis of the data collected.

Validity of the findings

The Conclusions section remains the same as before. The authors have just rearranged the words. This section should be strengthened with discussions that support the goals set out in the introduction. Unresolved gaps and questions should be clearly identified by the authors with suggestions and remedies for future directions in this field. The current state of affairs and established protocols should be discussed.

---

## Round 0.3 · accepted · Accept

Dear authors,

Many thanks for your hard-work. I am now accepting your manuscript for publication in PeerJ. Congratulations!

Reviewer 2 ·

Basic reporting

No comments

Experimental design

The authors have added Table 3 as suggested that enhances the review.

Validity of the findings

The conclusion section has been improved and the authors have addressed most of my concerns.